# DiffTAC: Temporal-Conditioned Latent Diffusion with Integrated Attention for Intermediate Frame Generation and Temporal Super-Resolution in Cardiac MRI

**Shilajit Banerjee**[1] (ID)                                    BANERJEE.SHILAJIT@TCS.COM
**Aniruddha Sinha**[1] (ID)                                              ANIRUDDHA.S@TCS.COM
[1] *Connected Digital Health, TCS Research, Kolkata, India*

**Editors:** Accepted for publication at MIDL 2026

## Abstract

Cardiac cine MRI captures dynamic cardiac motion, yet its temporal resolution remains fundamentally constrained by long acquisition times and breath-hold requirements. We introduce DiffTAC, a latent diffusion framework that synthesizes intermediate cardiac phases by treating time as an explicit conditioning variable. Using the end-diastolic (ED) and end-systolic (ES) frames as anatomical anchors, DiffTAC performs denoising in the latent space of a pretrained variational autoencoder and conditions generation on a learnable temporal embedding that specifies the desired phase location within the cardiac cycle. To effectively fuse temporal conditioning with anatomical context, we propose the Integrated Attention Block (IAB), a unified module that combines self-attention and cross-attention to modulate spatial features according to the target temporal position. This design enables the model to synthesize anatomically coherent, temporally smooth intermediate frames. Experiments on multiple publicly available datasets demonstrate that DiffTAC produces highly realistic intermediate phases and achieves superior temporal consistency compared to classical interpolation, optical-flow–based reconstruction, and ablated variants of our architecture. These findings show that modeling time as a conditioning signal within a diffusion framework provides an effective and acquisition-free solution for temporal super-resolution in cardiac MRI.

**Keywords:** Cardiac MRI, Frame Interpolation, Diffusion Models, Cross-Attention, Latent Space, Temporal Consistency

## 1. Introduction

Cardiac cine MRI provides dynamic information about ventricular contraction, relaxation, and myocardial motion, making it the reference standard for quantitative cardiac function assessment (Sechtem et al., 1987; Heitner et al., 2012; Vollbrecht et al., 2023). However, achieving high temporal resolution requires acquiring many cardiac phases, which increases scan duration and often exceeds what is feasible within a single breath-hold. As a result, clinical protocols typically acquire only a limited number of frames (e.g., 20–30 per cardiac cycle), reducing the temporal fidelity needed for fine-grained wall-motion analysis or strain estimation. High–frame-rate reconstructions obtained through post-processing can therefore enable improved strain estimation, motion analysis, and visualization of subtle physiological events that are not captured at standard frame rates. They may also support downstream applications such as fluid–structure interaction modeling and digital-twin cardiac simulations, where smooth and temporally continuous myocardial motion is essential.

A natural strategy to overcome this limitation is to synthesize intermediate cardiac phases after acquisition. Classical interpolation (e.g., linear, spline) cannot capture the highly non-linear deformation of the heart (Preechakul et al., 2022; Karras et al., 2018). Optical-flow-based methods are sensitive to noise and often fail during rapid contraction or valve motion (Shah and Xuezhi, 2021). Deep generative models hold promise, but pixel-space generators frequently struggle to maintain anatomical correctness or produce smooth temporal transitions across frames.

Diffusion models (Ho et al., 2020) have recently demonstrated exceptional generative fidelity and stability, especially in medical imaging (Kazerouni et al., 2023; Webber and Reader, 2024). Their iterative denoising formulation provides strong structural priors while avoiding mode collapse. Latent diffusion models (LDMs) (Rombach et al., 2021) operate in the latent space of a pretrained VAE. They offer computational efficiency and improved anatomical coherence—properties particularly desirable for cardiac cine MRI (Pinaya et al., 2022; Kim and Park, 2024).

In the context of cardiac motion, the end-diastolic (ED) and end-systolic (ES) frames represent two physiologically meaningful boundary conditions. These frames capture the extremes of ventricular relaxation and contraction and are readily available across standard clinical views. While ED and ES provide rich anatomical context, they do not specify when within the cardiac cycle an intermediate phase should occur. Thus, the central challenge in cardiac frame synthesis is to combine anatomical context from ED/ES with an explicit representation of time. Recent work explores diffusion and deep learning models for cardiac reconstruction and temporal enhancement. CaLID (Bubeck et al., 2025) uses latent diffusion to interpolate sparse slices for 3D cardiac reconstruction, but it does not use ED/ES anchors or treat time as an explicit conditioning signal. UVI-Net (Kim et al., 2024) adopts a learnable, flow-based interpolation strategy, sharing similar motion assumptions with classical optical-flow methods. TSSC-Net (Zhou et al., 2025) performs diffusion-driven temporal super-resolution for 4D MRI using start–end frame conditioning, but it focuses on volumetric motion and does not incorporate a learnable temporal embedding in latent space. Feature-based interpolation in DT-MRI (Yang et al., 2012) reconstructs diffusion tensor fields by interpolating eigenvalues and orientations, but its scope is limited to tensor data rather than cine intensity sequences.

Temporal super-resolution for 4D Flow MRI (Callmer et al., 2025) enhances velocity-field dynamics using residual networks, but it operates directly in pixel space and uses 3D convolutions which can be compute heavy. Building on these advances, we introduce DiffTAC, a diffusion-based temporal synthesis framework that treats time as an explicit conditioning variable. DiffTAC uses a learnable sinusoidal temporal embedding to encode the target phase position between ED and ES and injects this temporal information into a conditional U-Net (Ronneberger et al., 2015). To more effectively fuse temporal conditioning with ED/ES anatomy, we propose the Integrated Attention Block (IAB), a unified module that merges self-attention and cross-attention pathways, enabling the model to learn a smooth and physiologically plausible cardiac motion manifold.

We evaluate our method on the two publicly available datasets and conduct extensive analyses, including ablations and ground truth (GT) free temporal interpolation experiments. Results show that DiffTAC generates anatomically faithful frames, achieves su-

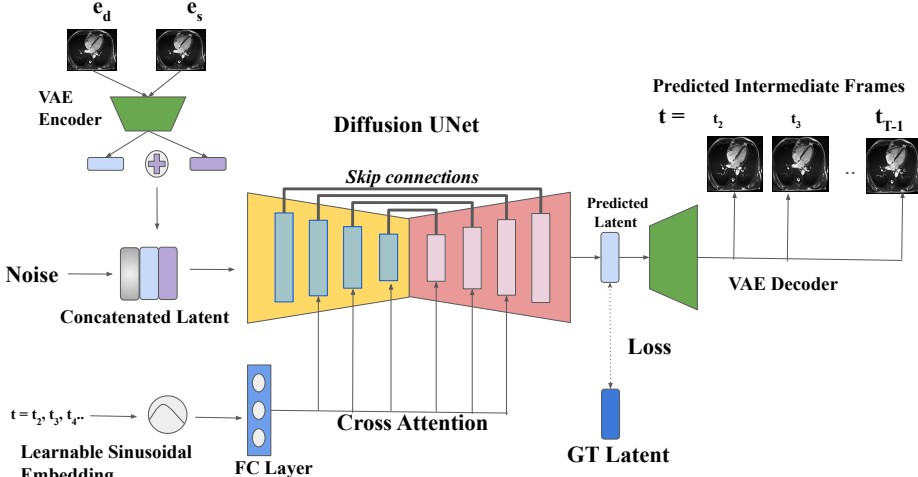

Figure 1: Overview of the DiffTAC framework. ED and ES frames are first encoded into the VAE latent space and concatenated with the noisy intermediate latent. A learnable sinusoidal embedding encodes the target temporal position and conditions the diffusion U-Net through cross-attention. The U-Net predicts the denoised latent, which is compared against the ground-truth latent during training. At inference, the predicted latent is decoded by the VAE to generate intermediate cardiac frames corresponding to any desired temporal position.

perior temporal smoothness, and extrapolates beyond observed phases, enabling temporal super-resolution without additional scan burden.

Our key contributions are:

1. We introduce DiffTAC, a diffusion-based framework that synthesizes cardiac frames between ED and ES by treating time as an explicit conditioning signal.

2. We propose the Integrated Attention Block (IAB), a novel self–cross attention module that effectively fuses temporal and anatomical information within a latent diffusion U-Net.

3. We demonstrate that DiffTAC enables robust temporal interpolation and supports cardiac temporal super-resolution in a fully data-driven manner.

4. We validate our approach on two publicly available datasets, showing consistent improvements over interpolation, optical-flow baselines, and ablated variants.

The rest of the paper is organized as follows. We present the details of the proposed method in Section 2. The details of the experiments and the results are given in Section 3. Finally, we conclude the paper in Section 6.

## 2. Methodology

### 2.1. Problem Formulation

We consider a full cardiac cycle represented as a cine sequence $\mathbf{X} = \{x_1, x_2, \ldots, x_T\}$, where $x_1$ and $x_T$ correspond to successive end–diastolic (ED) frames. The end–systolic (ES) frame

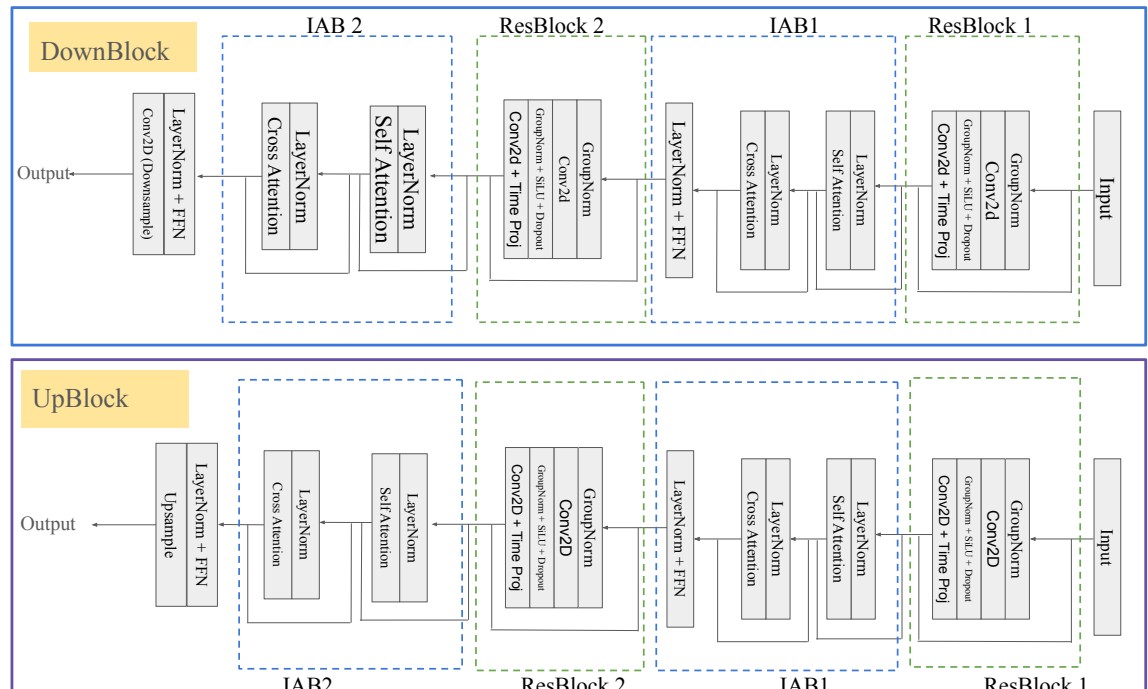

Figure 2: Proposed DownBlock and UpBlock structures in DiffTAC. Each block applies two Residual Blocks (ResBlock 1 and 2), each followed by an Integrated Attention Block (IAB1 and IAB2), which merges self-attention, cross-attention, and Feed forward Network (FFN) within a residual pathway. DownBlocks conclude with Conv2D downsampling, while UpBlocks fuse encoder skip connections and apply upsampling. Together, these modules integrate ED/ES anatomical context with temporal conditioning across spatial scales.

appears at a dataset–dependent position within the cycle, but the temporal evolution from ED to ES and from ES back to the next ED is not symmetric. All frames are uniformly sampled in time, and the complete cycle is mapped to a normalized temporal coordinate $\tau \in [0, 1]$, such that $\tau = 0$ and $\tau = 1$ both correspond to ED, while ES occurs at some intermediate $\tau = \tau_{\mathrm{ES}}$.

For a full cardiac cycle, DiffTAC is trained on both motion segments. Using the provided ED and ES annotations, we construct (i) an ED→ES interval, where ED is the start frame and ES is the end frame, and the model learns to generate the intermediate ES-bound phases; and (ii) an ES→ED interval, where ES becomes the start frame and the subsequent frames toward the next ED define the target positions. Thus, each full cycle contributes two supervised temporal trajectories, allowing the model to learn intermediate-frame generation consistently across both directions of cardiac motion.

## 2.2. Latent Space Encoding

We operate in the latent space of a pretrained variational autoencoder (VAE), adapted from Stable Diffusion (Rombach et al., 2021) and fine-tuned on cardiac MRI. The VAE maps images of size $H \times W \times C_{\mathrm{img}}$ to latent tensors of size $\frac{H}{8} \times \frac{W}{8} \times C_\ell$, providing an $8\times$ spatial

reduction while preserving anatomical structure. Here $H$, $W$ and $C$ represents the height, width and channels respectively. The encoder $\mathcal{E}$ outputs a diagonal Gaussian distribution from which a latent code is sampled:

$$z = \mathcal{E}(x), \qquad \mathcal{E}(x) \sim \mathcal{N}\big(\mu_E(x),\, \sigma_E^2(x)\big). \tag{1}$$

The decoder maps latents back to the image domain, $\hat{x} = \mathcal{D}(z)$. VAE weights remain frozen during training. Latents for the end-diastolic (ED) and end-systolic (ES) frames are $z_{\text{ed}} = \mathcal{E}(x_0)$ and $z_{\text{es}} = \mathcal{E}(x_{\tau_{\text{ES}}})$, which serve as anatomical anchors. The diffusion model predicts an intermediate latent $\hat{z}_\tau$ for any temporal position $\tau \in [1, 0]$, decoded as $\hat{x}_\tau = \mathcal{D}(\hat{z}_\tau)$.

### 2.3. Model Architecture

The proposed model is a conditional latent diffusion U-Net that predicts the noise $\epsilon_\theta(x_t, t, c_\tau)$ added to a latent variable during the forward diffusion process. The network input $x_t \in \mathbb{R}^{3C_\ell \times (H/8) \times (W/8)}$ is formed by concatenating three $C_\ell$-channel latent tensors: the noisy latent $z_t$, the ED latent $z_{\text{ed}}$, and the ES latent $z_{\text{es}}$. The diffusion timestep is denoted by $t$, and the temporal conditioning token $c_\tau \in \mathbb{R}^{1 \times d_c}$ is derived from the target phase position $\tau$. The overall block diagram is shown in Fig 1.

**Overall structure.** The U-Net follows a four-level encoder–decoder architecture with channel widths $\{C, 2C, 4C, 8C\}$, where $C$ denotes the initial number of channels. Each resolution stage contains two residual blocks followed by two *Integrated Attention Blocks* (IAB1, IAB2), as illustrated in Fig. 2. Skip connections propagate encoder features to the corresponding decoder stages. Downsampling is performed using a stride-2 $3 \times 3$ convolution, and upsampling uses nearest-neighbor interpolation followed by a $3 \times 3$ convolution.

**Residual blocks.** Each residual block processes a feature map $h \in \mathbb{R}^{B \times C \times H \times W}$ using a standard Group Normalization (GN)–SiLU–Conv2d stack. Timestep conditioning is injected by adding a projected timestep embedding $W_t e_t$ to the intermediate activation, giving

$$u = \text{Conv}(\text{SiLU}(\text{GN}(h))) + W_t e_t,$$

where $e_t$ is the diffusion timestep embedding. If the input and output channel dimensions differ, an optional $1 \times 1$ convolution is applied to the skip path so that the residual connection can be added consistently.

**Temporal conditioning.** The target temporal position $\tau$ is encoded using a sinusoidal *positional encoding* (PE) (Vaswani et al., 2017), adapted to continuous time and modulated through learnable affine parameters. A base positional encoding is first computed from the discretized temporal index $p(\tau)$ (see Appendix A):

$$\phi(\tau) = \text{clip}(s) \odot \text{PE}(p(\tau)) + b, \tag{2}$$

where $s, b \in \mathbb{R}^d$ are learnable scale and shift vectors. The modulated embedding is then projected to a compact conditioning token $c_\tau = W_{\text{proj}}\, \phi(\tau)$ which encodes the desired temporal location and conditions all cross-attention layers inside the IAB modules.

**Integrated Attention Block (IAB).** Each IAB module fuses spatial features with temporal conditioning using multi-head self-attention (SA) followed by multi-head cross-attention (CA). Given flattened spatial tokens $Z \in \mathbb{R}^{(HW) \times d}$, the SA operation is

$$\mathrm{SA}(Z) = \mathrm{softmax}\left( \frac{ZW_Q(ZW_K)^\top}{\sqrt{d_k}} \right) ZW_V. \tag{3}$$

Cross-attention uses queries from $Z$ and keys/values from the temporal token $c_\tau$:

$$\mathrm{CA}(Z, c_\tau) = \mathrm{softmax}\left( \frac{ZW_Q(c_\tau W_{K_c})^\top}{\sqrt{d_k}} \right) c_\tau W_{V_c}. \tag{4}$$

With residual connections and LayerNorm (LN), the IAB update, consisting of SA and CA, is followed by a single position-wise feed-forward network (FFN), as shown below:

$$\begin{aligned} \mathbf{Y}_1 &= \mathbf{Z} + \mathrm{SA}(\mathrm{LN}(\mathbf{Z})), \\ \mathbf{Y}_2 &= \mathbf{Y}_1 + \mathrm{CA}(\mathrm{LN}(\mathbf{Y}_1), \mathbf{c}_\tau), \\ \mathbf{Z}' &= \mathbf{Y}_2 + \mathrm{FFN}(\mathrm{LN}(\mathbf{Y}_2)). \end{aligned} \tag{5}$$

**Decoder.** The decoder mirrors the encoder and receives concatenated skip features at each resolution. Each stage applies three residual blocks and two IAB modules, allowing temporal conditioning to refine features even at fine spatial scales. Upsampling is performed by nearest-neighbor interpolation followed by a $3 \times 3$ convolution.

**Output layer.** The final decoder feature map is normalized and activated as $h_{\mathrm{out}} = \mathrm{SiLU}(\mathrm{GN}(h))$, and then mapped through a $3 \times 3$ convolution to predict the latent-space noise $\hat{\epsilon} = \mathrm{Conv}_{3\times3}(h_{\mathrm{out}})$. This prediction $\hat{\epsilon}$ is used by the reverse diffusion process to reconstruct a temporally conditioned latent frame corresponding to $\tau$.

## 2.4. Training Procedure

We train the model to predict the diffusion noise added to latent frames conditioned on the end-diastolic and end-systolic latents and a temporal embedding. Let $z_{\mathrm{ed}}, z_{\mathrm{es}} \in \frac{H}{8} \times \frac{W}{8} \times C$ denote the latent representations of the boundary frames, where $B$ is the batch size.

2.4.1. TRAINING SETUP

At each iteration, we sample a temporal position $\tau \in [1, 0]$ between end-diastole and end-systole and compute its learnable sinusoidal embedding $\phi(\tau) \in \mathbb{R}^d$ and project it through a small fully connected layer *outside* the diffusion U-Net to form the conditioning token:

$$c_\tau = W_{\mathrm{proj}} \phi(\tau) \in \mathbb{R}^{1 \times d_c}. \tag{6}$$

This projection prepares the temporal embedding for injection into the U-Net via the cross-attention layers. Given the ground-truth latent $z_\tau$ at phase $\tau$, the forward diffusion step is

$$z_t = \sqrt{\bar{\alpha}_t}\, z_\tau + \sqrt{1 - \bar{\alpha}_t}\, \epsilon, \qquad \epsilon \sim \mathcal{N}(0, I). \tag{7}$$

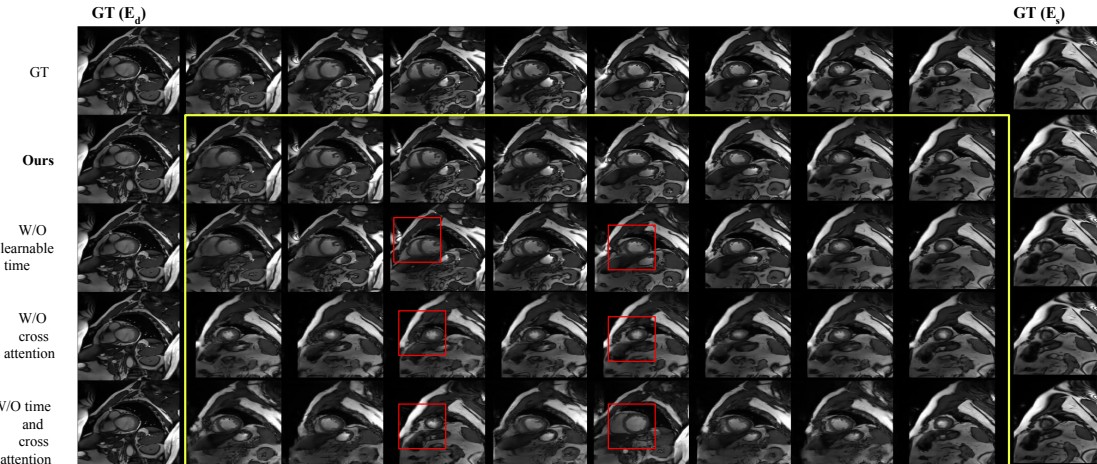

Figure 3: Qualitative results on the ACDC dataset showing intermediate cardiac frame synthesis from ED to ES. The yellow box contains all generated intermediate frames for each method. Diff-TAC closely matches ground-truth dynamics and preserves ventricular geometry throughout the cardiac cycle. In contrast, the ablated variants—without the learnable temporal embedding, without cross-attention, or without both—show motion inconsistencies and anatomical distortions, highlighted in the red boxes.

where $t$ is the diffusion timestep and $\bar{\alpha}_t$ is the cumulative noise schedule. The U-Net predicts the added noise conditioned on the noisy latent and the ED/ES boundary frames:

$$\hat{\epsilon} = \text{U-Net}\big([z_t, z_{\text{ed}}, z_{\text{es}}], \ t, \ c_\tau\big). \tag{8}$$

The timestep embedding encodes $t$, while the token $c_\tau$ provides temporal conditioning at all attention layers. Training minimizes the mean squared error between predicted and true noise:

$$\mathcal{L} = \mathbb{E}_{z_\tau, \epsilon, \tau, t}\left[\|\epsilon - \hat{\epsilon}\|_2^2\right]. \tag{9}$$

This loss guides the model to denoise latent frames according to both the anatomy and the target temporal phase, enabling smooth intermediate-frame synthesis during inference.

## 3. Experiments and Results

### 3.1. Datasets

We evaluate DiffTAC on two public cine MRI datasets: the Sunnybrook Cardiac Data (SCD) (Radau et al., 2009) and ACDC (Bernard et al., 2018). Both provide short-axis cine sequences with annotated end-diastolic (ED) and end-systolic (ES) frames. SCD contains 45 subjects spanning healthy and pathological cases and provides *full* cardiac cycles (ED→ES→ED) with roughly 20 frames acquired at 1.5T and an in-plane resolution of 1.25–1.5 mm at $256 \times 256$ resolution. In contrast, ACDC includes 150 subjects across five diagnostic categories (NOR, MINF, DCM, HCM, RV) and provides only the *ED→ES* half-cycle with 10 standardized frames acquired on 1.5T or 3T scanners at 1.37–1.68 mm

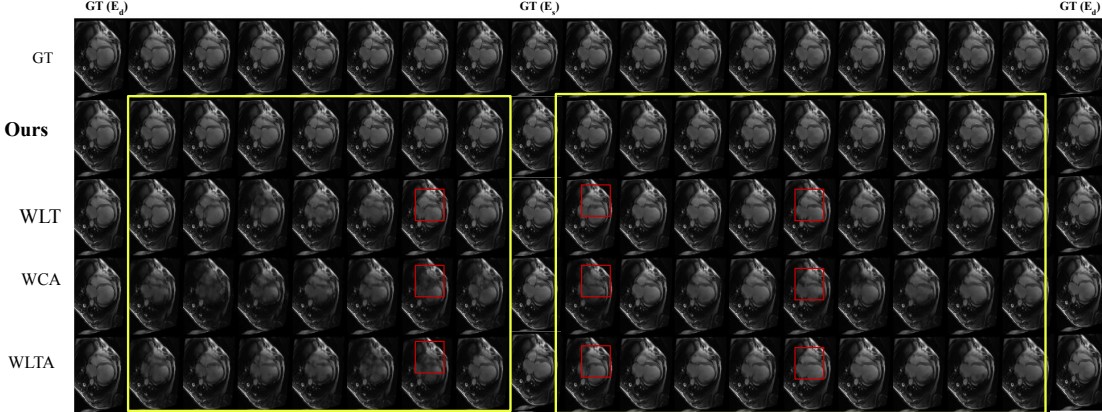

Figure 4: Qualitative results on the Sunnybrook dataset showing cardiac motion from ED to ES. The yellow boxes contain all generated intermediate frames for each method. DiffTAC (Ours) produces temporally smooth and anatomically consistent transitions that align well with ground-truth dynamics. In contrast, the ablated variants—without the learnable temporal embedding (WLT), without cross-attention (WCA), or without both (WLTA)—show motion artifacts and structural distortions, which are marked with red boxes.

Table 1: Intermediate frame generation performance on the Sunnybrook and ACDC datasets. Higher PSNR/SSIM and lower LPIPS indicate better reconstruction quality.

| Dataset | Method | PSNR↑ | SSIM↑ | LPIPS↓ |
|---|---|---|---|---|
| Sunnybrook | Ours | **39.660** | **0.922** | **0.011** |
| | Linear | 33.450 | 0.897 | 0.125 |
| | Spline | 34.850 | 0.901 | 0.101 |
| | Optical Flow | 29.500 | 0.860 | 0.172 |
| ACDC | Ours | **33.950** | **0.898** | **0.110** |
| | Linear | 18.280 | 0.611 | 0.249 |
| | Spline | 18.600 | 0.617 | 0.242 |
| | Optical Flow | 16.690 | 0.569 | 0.335 |

resolution. All frames are center-cropped, resized to $256 \times 256$, and normalized to $[-1, 1]$. We use an 80/20 subject-level train–test split for both datasets.

### 3.2. Implementation Details

All images are resized to $256 \times 256$ and normalized to $[-1, 1]$. A pretrained Stable-Diffusion VAE encodes each frame into a $32 \times 32 \times 4$ latent; the VAE remains frozen throughout training. We train the diffusion model for 500 epochs using AdamW (learning rate $1 \times 10^{-4}$, weight decay $10^{-5}$), cosine learning-rate scheduling with 500 warmup steps, batch size 1, and gradient-accumulation steps 4. The noise process uses 1000 diffusion timesteps with scaled-linear $\beta$ scheduling. The learnable sinusoidal embedding table length is set to 200

Table 2: Comparison with learning-based temporal reconstruction methods on the Sunnybrook and ACDC datasets. Higher PSNR/SSIM and lower LPIPS indicate better performance. **Temporal SR** denotes the ability to generate cine sequences at denser temporal positions (higher frame rate) within the cardiac cycle.

| Dataset | Method | PSNR↑ | SSIM↑ | LPIPS↓ | Temporal SR |
|---------|--------|-------|-------|--------|-------------|
| | UVI-Net (Kim et al., 2024) | 32.313 | 0.896 | 0.121 | No |
| | TSSC-Net (Zhou et al., 2025) | 28.767 | 0.874 | 0.132 | 6× |
| Sunnybrook | Ours | **39.660** | **0.922** | **0.011** | **Flexible (2×, 3×, 4×, ...)** |
| | UVI-Net (Kim et al., 2024) | **35.091** | **0.925** | **0.035** | No |
| | TSSC-Net (Zhou et al., 2025) | 34.677 | 0.917 | 0.040 | 6× |
| ACDC | Ours | 33.950 | 0.898 | 0.110 | **Flexible (2×, 3×, 4×, ...)** |

Table 3: Ablation study on the Sunnybrook and ACDC datasets. Higher PSNR/SSIM and lower LPIPS indicate better reconstruction quality.

| Dataset | Configuration | PSNR↑ | SSIM↑ | LPIPS↓ |
|---------|---------------|-------|-------|--------|
| | Full Model | **39.660** | **0.922** | **0.011** |
| Sunnybrook | w/o Learnable Time Embedding | 34.105 | 0.900 | 0.116 |
| | w/o Cross-Attention | 28.740 | 0.866 | 0.167 |
| | w/o Time & Cross-Attention | 20.680 | 0.811 | 0.205 |
| | Full Model | **33.950** | **0.898** | **0.110** |
| ACDC | w/o Learnable Time Embedding | 24.640 | 0.850 | 0.181 |
| | w/o Cross-Attention | 21.400 | 0.845 | 0.195 |
| | w/o Time & Cross-Attention | 16.880 | 0.589 | 0.274 |

and the cross-attention conditioning dimension to 768. During inference, we employ DDIM sampling with 100 denoising steps. For any target temporal position $\tau \in [1, 0]$, the model predicts the corresponding latent $\hat{z}_\tau$ conditioned on the ED and ES latents $(z_{\text{ed}}, z_{\text{es}})$ and the time token $c_\tau$. The final reconstruction is obtained via the frozen VAE decoder.

### 3.3. Evaluation Metrics

We evaluate interpolation quality using four metrics. **PSNR** measures pixel-wise reconstruction accuracy. **SSIM** measures structural similarity. **LPIPS** (Zhang et al., 2018) measures perceptual similarity using deep features; lower is better. These metrics assess fidelity, perceptual quality, and temporal stability. As shown in Table 1, DiffTAC achieves strong performance across all metrics. Latent-space denoising preserves anatomical detail, and the temporal embedding with cross-attention improves phase alignment and global coherence.

### 3.4. Quantitative Results

Table 1 summarizes interpolation performance on the Sunnybrook and ACDC datasets. DiffTAC achieves the highest PSNR and SSIM and the lowest LPIPS on both datasets. Linear and spline interpolation offer limited improvements because they operate directly

Table 4: Ablation study on temporally interpolated frames (2× temporal super-resolution). Lower FID indicates better realism.

| Configuration | Sunnybrook FID↓ | ACDC FID↓ |
|---|---|---|
| Full Model | **23.060** | **60.170** |
| w/o Learnable Time Embedding | 40.770 | 90.610 |
| w/o Cross-Attention | 67.970 | 152.870 |
| w/o Time & Cross-Attention | 50.120 | 137.940 |

on pixel intensities and cannot model nonlinear cardiac deformation. Optical flow performs worse due to unreliable motion estimates in regions with rapid contraction or low contrast.

DiffTAC outperforms all baselines because it models cardiac motion as a smooth trajectory in latent space. The ED and ES latents provide anatomical anchors, and the temporal embedding steers the diffusion process toward the correct phase. This combination yields sharper structures, better texture reconstruction, and smoother temporal transitions, as also seen qualitatively in Figures 3 and 4. For more visual results see Appendix C.

### 3.5. Comparison with State of the Art

Table 2 compares DiffTAC with recent learning-based temporal reconstruction methods. We include UVI-Net (Kim et al., 2024), a learnable flow-based volumetric interpolation method, and TSSC-Net (Zhou et al., 2025) , a diffusion-based temporal super-resolution approach that can achieve fixed 6× upsampling using start and end frames. On the Sunnybrook dataset, which contains longer cardiac sequences with a larger number of intermediate frames, DiffTAC substantially outperforms both methods across all metrics, achieving higher PSNR and SSIM and markedly lower LPIPS, indicating more accurate and perceptually consistent reconstructions. On the ACDC dataset, UVI-Net attains higher image-based metrics, while DiffTAC performs phase-conditioned synthesis with flexible temporal scaling. Unlike TSSC-Net, which is limited to a fixed temporal factor, DiffTAC supports arbitrary temporal super-resolution within a unified framework. For more results on temporal super resolution of DiffTAC please refer to Apendix B.

### 3.6. Ablation Studies

We evaluate the impact of the learnable temporal embedding and cross-attention in Table 3. Removing the temporal embedding reduces the model's ability to identify the target phase, decreasing reconstruction accuracy and temporal smoothness. Removing cross-attention prevents effective fusion of temporal information with spatial features, leading to blurrier structures and weaker perceptual quality. Disabling both components produces the largest degradation. The qualitative effects are visible in Figures 3 and 4, where ablated variants exhibit motion artifacts and anatomical inconsistencies. These results show that DiffTAC benefits from three elements: latent-space denoising, explicit temporal conditioning, and the Integrated Attention Block for multi-scale fusion (see Appendix C for more).

### 3.7. Temporal Super-Resolution

Beyond interpolation, DiffTAC performs temporal super-resolution by generating cardiac phases within the acquired ED-ES interval. We evaluate $2\times$, $3\times$, $4\times$, and $5\times$ temporal super-resolution by sampling the temporal position $\tau \in [1, 0]$ at increasingly dense intervals (e.g., $\tau = \{0.10, 0.15, 0.20, \ldots\}$ for higher upsampling factors). As the upsampling factor increases, the model synthesizes more intermediate frames, and we observe a gradual reduction in visual quality, consistent with the increasing difficulty of high-ratio in-between frame synthesis. Since no ground-truth frames exist at these densified temporal positions, we assess realism using the Fréchet Inception Distance (FID) (Yu et al., 2021), as shown in Table 4. DiffTAC achieves the lowest FID across both datasets. Removing the learnable temporal embedding or cross-attention leads to higher FID, indicating the importance of these components when no ground truth is available. These results show that DiffTAC provides reliable temporal super-resolution over a range of upsampling factors and can generate higher-frame-rate cine sequences without additional MRI acquisition. For more results see Appendix B.

## 4. Discussion

The learnable sinusoidal temporal embedding provides a continuous, smooth representation of the target phase $\tau$, which is essential for modeling cardiac motion. Standard positional encodings offer a fixed Fourier basis, but the learnable variant adapts this basis to the training distribution, allowing the network to capture dataset-specific temporal progression. Because cardiac motion evolves smoothly and follows a cyclic pattern, representing $\tau$ through a mixture of learnable sinusoidal components creates an implicit temporal manifold on which intermediate phases lie. The projection $c_\tau = W_{\mathrm{proj}}\phi(\tau)$ then maps this trajectory into the conditioning space, enabling the diffusion model to treat time as a continuous control signal rather than a discrete index.

During training, the end-diastolic (ED) and end-systolic (ES) frames are not used as supervision targets; instead, they are encoded using a frozen VAE and concatenated with the noisy latent of the target intermediate frame as conditioning inputs to the diffusion U-Net. This allows the model to learn view- and anatomy-specific feature representations, while supervision is applied only through noise prediction on the intermediate latent. Conditioning on ED and ES constrains generation to the correct anatomical feature space, which is important because cardiac MRI can be acquired from different views (e.g., 2-chamber, 4-chamber, short-axis). By providing ED and ES latents, the model is guided to synthesize cine sequences that remain consistent with the given input anatomy and imaging plane.

The Integrated Attention Block (IAB) formalizes how temporal information modulates spatial representations. In diffusion models, self-attention alone captures spatial correlations, but it cannot enforce temporal alignment between ED and ES. Cross-attention provides a mechanism for injecting the temporal embedding into each spatial location. By using the same queries as self-attention but replacing keys and values with the temporal token, the model learns a linear operator that selects temporal features relevant to each spatial region. This gives a principled way to condition the denoising trajectory on the desired phase. The residual combination of self-attention, cross-attention, and feed-forward

updates forms a shallow approximation to a dynamical system in which spatial features evolve under a temporally modulated flow field.

Together, the learnable temporal embedding and IAB structure allow the network to encode temporal position as a continuous latent variable and to apply it consistently across all spatial scales. This yields a model that is sensitive to temporal ordering while remaining robust to anatomical variation, enabling coherent intermediate-frame generation and higher-order temporal super-resolution.

## 5. Potential Applications

The ability of DiffTAC to generate temporally super-resolved cardiac phases within the acquired ED–ES interval enables several practical applications in cardiac imaging and computational modeling. First, the method supports high–frame-rate cine reconstruction, producing temporally dense image sequences without requiring longer or repeated MRI acquisitions. Such high-resolution temporal data can improve clinical tasks including myocardial strain estimation (Amzulescu et al., 2019), regional wall-motion assessment (Wahba et al., 2001), detection of subtle functional abnormalities, and enhanced visualization of rapid physiological events that may not be captured at standard frame rates.

Second, DiffTAC can facilitate fluid–structure interaction (FSI) (Kock et al., 2008) studies by providing smooth and temporally continuous myocardial boundary motion. This enables more accurate numerical simulations of ventricular hemodynamics, leading to improved characterization of blood flow patterns and pressure fields across the cardiac cycle.

Third, the framework is useful for digital-twin cardiac modeling (Zhao et al., 2025), where patient-specific dynamic sequences are required to calibrate or validate personalized biophysical models. By generating anatomically consistent intermediate phases, DiffTAC helps fill temporal gaps in sparsely sampled cine data.

Finally, the ability to synthesize physiologically realistic motion trajectories (Liu et al., 2024) can support downstream tasks such as motion correction, data augmentation, temporal harmonization across heterogeneous datasets, and the creation of dense training sequences for models requiring fine-grained temporal supervision.

## 6. Conclusion

We present DiffTAC, a latent diffusion framework for cardiac cine interpolation and temporal super-resolution. By treating time as an explicit conditioning variable and combining ED/ES anatomical context with the proposed Integrated Attention Block, the model generates smooth and anatomically consistent intermediate frames across the cardiac cycle. Experiments on the Sunnybrook and ACDC datasets show strong improvements over interpolation and optical-flow baselines, as well as clear benefits from our design choices through ablation studies. DiffTAC also extrapolates beyond the acquired range, enabling flexible temporal super-resolution without additional scan burden. These results suggest that diffusion models with explicit temporal conditioning offer a promising direction for reconstructing high-frame-rate cardiac cine MRI in clinically constrained acquisition settings.

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

## Appendix A. Mathematical basis of the Learnable Sinusoidal Time Embedding

Let $\tau \in [0, 1]$ denote a normalized temporal position. We map $\tau$ to a discrete table index via

$$p(\tau) = \lfloor \tau(L-1) \rfloor, \qquad p(\tau) \in \{0, \ldots, L-1\}, \tag{10}$$

where $L$ is the embedding-table length. The corresponding base sinusoidal embedding $\mathrm{PE}(p) \in \mathbb{R}^d$ is defined componentwise as

$$\mathrm{PE}_{2i}(p) = \sin(p\,\omega_i), \qquad \mathrm{PE}_{2i+1}(p) = \cos(p\,\omega_i), \qquad \omega_i = 10000^{-2i/d}. \tag{11}$$

This provides a smooth Fourier-like basis over the discrete index $p$. To adapt the embedding to dataset-specific temporal dynamics, we apply an affine modulation:

$$\phi(\tau) = \mathrm{clip}(s) \odot \mathrm{PE}\big(p(\tau)\big) + b, \tag{12}$$

where $s, b \in \mathbb{R}^d$ are learnable scale and shift parameters, and $\odot$ denotes element-wise multiplication. The clamping of $s$ improves numerical stability and controls the amplitude of high-frequency components.

In our setting, a cine sequence consists of $T$ ordered frames $\{x_1, \ldots, x_T\}$ with annotated end-diastolic (ED) and end-systolic (ES) frames. For Sunnybrook, the full cardiac cycle ED→ES→ED is available, so we normalize temporal positions such that the first ED frame maps to $\tau = 0$ and the final ED frame maps to $\tau = 1$. The ES frame naturally lies at some intermediate $\tau \in [1, 0]$ depending on its frame index. For ACDC, only an ED→ES half-cycle

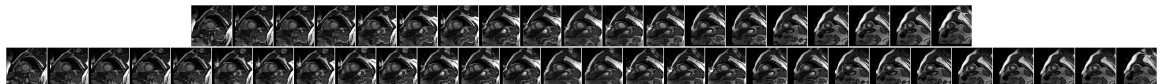

Figure 5: Temporal super resolution on cardiac sequences on the ACDC dataset. The top row presents 2× temporal super-resolution, and the bottom row presents 3× temporal super-resolution, showing denser in-between frames within the acquired ED–ES interval. Due to the large number of generated frames, images are shown at a reduced scale; please zoom in to view structural details clearly.

Figure 6: Temporal super resolution on cardiac sequences on the Sunnybrook dataset. The top row shows 2× temporal super-resolution, and the bottom row shows 3× temporal super-resolution, both producing denser in-between frames within the acquired ED–ES interval. Because the figure contains many generated frames, images are displayed at reduced size; please zoom in for detailed visualization.

is provided; therefore ED is mapped to $\tau = 0$ and ES to $\tau = 1$. All intermediate frames occupy evenly spaced $\tau$ values in their respective ranges, while temporal super-resolution is performed by sampling $\tau$ more densely within $[1, 0]$.

The resulting modulated embedding $\phi(\tau)$ is finally projected to

$$c_\tau = W_{\text{proj}}\phi(\tau), \tag{13}$$

which serves as the temporal conditioning token for cross-attention within the IAB modules. This construction admits several useful properties. (i) *Local continuity*: adjacent values of $\tau$ map to nearby indices $p(\tau)$, and the sinusoidal basis varies smoothly with $p$, so $c(\tau)$ is locally continuous in $\tau$ (modulo discretization). (ii) *Expressivity*: the sinusoidal basis provides a set of frequency components (Fourier features), and the learnable affine parameters allow the model to reweight and shift these components to emphasize dataset-specific temporal patterns. (iii) *Generalization*: because the base uses sinusoidal components rather than arbitrary lookup vectors, the embedding extrapolates in a structured way beyond seen indices, giving stable behavior for dense sampling of $\tau$ during temporal super-resolution.

Gradients backpropagate through $s$ and $b$ directly. Since the dataset provides a fixed set of temporal positions (the eight interior frames between ED and ES), $\tau$ takes only these discrete values during training. Thus the mapping $p(\tau)$ is effectively fixed for all training examples, and the learnable affine parameters $s$ and $b$ adapt to these known temporal indices. During inference, however, we evaluate the embedding at dense values of $\tau$ (e.g. $\tau = 0.05, 0.10, 0.15, 0.20, 0.25, \ldots$) for temporal super-resolution. The sinusoidal basis ensures that these unseen positions still produce smooth, structured embeddings, while the learned affine modulation preserves dataset-adapted temporal trends. Together, this yields a compact, stable representation for providing continuous temporal conditioning to the cross-attention layers in the IAB modules.

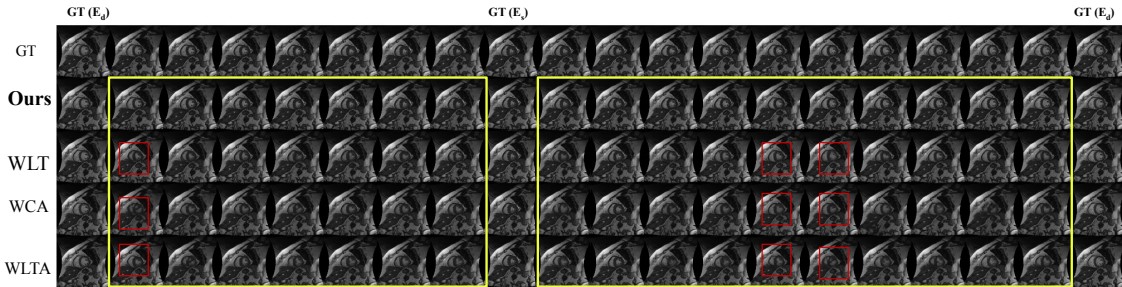

Figure 7: Supplementary qualitative results on the Sunnybrook dataset. Generated frames (yellow box) illustrate the temporal progression within the ED-ES interval. DiffTAC maintains smooth motion and structural consistency, whereas models without temporal embedding (WLT), without cross-attention (WCA), or without both (WLTA) show irregular transitions and reduced stability (red markers).

Table 5: **FID scores for temporal super resolution at increasing super-resolution factors.** We evaluate $2\times$, $3\times$, $4\times$, and $5\times$ temporal super-resolution on the Sunnybrook and ACDC datasets. Lower FID indicates closer alignment to the distribution of real cardiac cine MRI images.

| Dataset | $2\times$ | $3\times$ | $4\times$ | $5\times$ |
|---|---|---|---|---|
| Sunnybrook | 27.32 | 32.58 | 34.42 | 37.22 |
| ACDC | 62.33 | 66.18 | 67.24 | 71.24 |

## Appendix B. Experiments on Temporal Super-Resolution

In addition to the $2\times$ super-resolution experiments reported in the main manuscript, we further evaluate DiffTAC under $3\times$, $4\times$, and $5\times$ temporal super resolution. For each scaling factor, the temporal embedding $\tau$ is sampled more densely within the interval $[1, 0]$, and the model generates the corresponding intermediate frames. Because ground-truth images do not exist for these extrapolated positions, we assess realism using the Fréchet Inception Distance (FID). As shown in Table 5, FID increases gradually with the interpolation factor, which reflects the increasing difficulty of predicting frames farther from the ED–ES boundary frames. Even so, DiffTAC preserves stable performance across all settings.

Figures 5 and 6 provide visual examples of $2\times$ and $3\times$ temporal super resolution on the ACDC and Sunnybrook datasets, respectively. The generated sequences display smooth temporal evolution and coherent anatomical structure. Due to the large number of frames, images are shown at reduced scale, and we recommend zooming in for detailed inspection. These examples demonstrate that DiffTAC can synthesize plausible long-range temporal trajectories, enabling high-frame-rate cine reconstruction without requiring additional MRI acquisition.

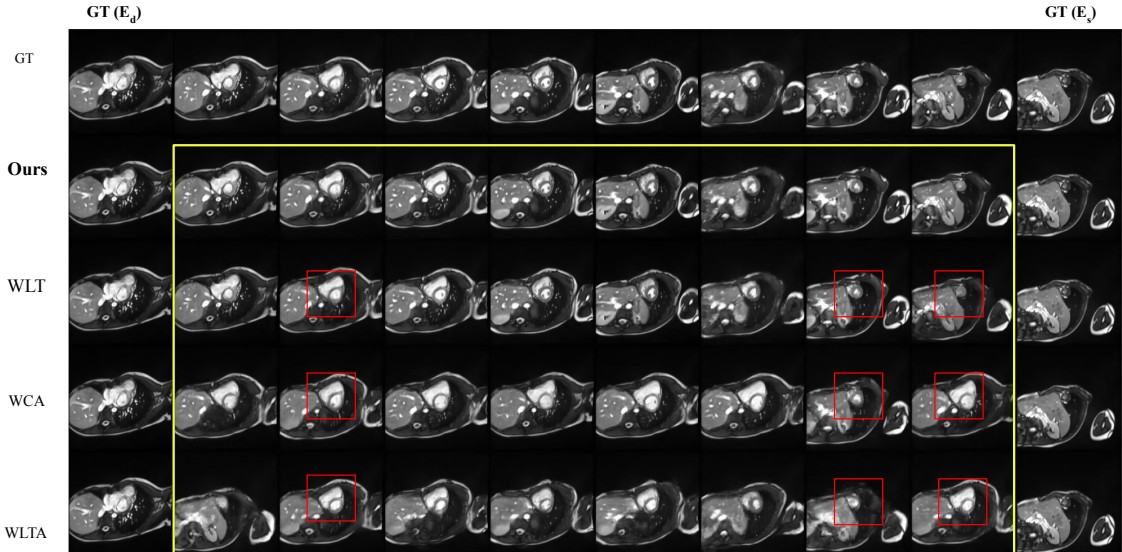

Figure 8: Additional qualitative examples from the ACDC dataset. The yellow box highlights all generated intermediate frames. DiffTAC produces coherent temporal transitions across the cardiac cycle, whereas the ablated variants exhibit less stable evolution of cardiac structures and reduced temporal smoothness, indicated by red markers.

## Appendix C. Additional Visual Results

In this section, we provide extended qualitative examples for both datasets used in our study. Figure 7 shows additional intermediate-frame generation results on the Sunnybrook dataset, illustrating how DiffTAC produces smooth and coherent transitions across the ED–ES interval. Figure 8 presents corresponding results on the ACDC dataset, where the model maintains anatomical consistency and temporal continuity across all generated phases. These examples further highlight the effect of removing temporal conditioning or cross-attention, as the ablated variants exhibit visible disruptions in motion progression and spatial alignment.

## Appendix D. Limitations

While DiffTAC shows strong performance in both interpolation and temporal super-resolution, several practical considerations remain. The model uses the ED and ES frames as boundary anchors, which is a clinically realistic setting but may limit performance if these frames are severely corrupted. Nevertheless, this dependency is far milder than methods requiring full-frame inputs, segmentation masks, or optical-flow estimations.

Temporal super resolution performance decreases moderately as the super-resolution factor increases, as reflected by rising FID values in Table 5. This behavior is expected: the model is asked to synthesize motion increasingly distant from any acquired data. Importantly, DiffTAC still maintains consistent anatomical structure and temporal smoothness even at higher factors, unlike traditional interpolation or motion-based methods that de-

grade rapidly in the same regime. Finally, although diffusion models are computationally heavier than direct neural interpolators, operating in latent space significantly reduces cost and makes inference practical for cine MRI. Given the substantial gains in temporal coherence and realism, this trade-off is favorable for applications that require high-quality dynamic reconstruction.

