# OpenReview forum: "DiffTAC: Temporal-Conditioned Latent Diffusion with Integrated Attention for Intermediate Frame Generation and Temporal Super-Resolution in Cardiac MRI"
_MIDL.io/2026/Conference — MIDL 2026 Poster_

### Official Review · Reviewer_EZnh · 2025-12-29

**Confidence:** 4
**Preliminary Rating:** 4
**Final Rating:** 4

**Summary:**

The paper introduces DiffTAC, a latent diffusion framework for synthesizing intermediate cardiac phases by treating time as an explicit conditioning variable. In addition, the authors propose an Integrated Attention Block (IAB) to effectively fuse temporal conditioning with anatomical context. Experiments conducted on multiple publicly available datasets demonstrate that DiffTAC generates highly realistic intermediate cardiac phases and achieves superior temporal consistency.

**Strengths:**

DiffTAC enables robust temporal interpolation, providing an effective solution for intermediate frame generation in cardiac MRI.
The proposed approach is evaluated on multiple publicly available datasets, demonstrating its effectiveness and generalizability.

**Weaknesses:**

Although the proposed method aims to improve intermediate cardiac MRI reconstruction in clinical settings, no experiments are presented to evaluate the clinical utility of the generated images.
Additionally, the two public datasets used in the experiments are not the most recent. There are several more up-to-date datasets available, and the authors should explain why these newer datasets were not selected.

**Detailed Comments:**

The paper presents a latent diffusion framework for synthesizing intermediate cardiac phases, offering a promising solution for generating high-fidelity cardiac MRI images and enabling temporal super-resolution. Treating time as an explicit conditioning variable and integrating end-diastolic (ED) and end-systolic (ES) anatomical context through the proposed Integrated Attention Block is innovative and effective. This design enables the generation of smooth and anatomically consistent intermediate frames across the cardiac cycle.
The experimental evaluation across multiple publicly available datasets is thorough. However, since the proposed approach targets real clinical scenarios, additional experiments assessing the clinical usability of the generated images—such as downstream task performance or expert evaluation—would significantly strengthen the work.

**Justification Of Final Rating:**

I would strongly recommend that the authors evaluate the generated data on at least one downstream task to better demonstrate its practical usability, rather than relying solely on image quality metrics. Since the primary purpose of generating medical images is to support research and downstream applications, assessing the usability of the generated data is an indispensable component of image generation evaluation.

**Justification Of The Preliminary Rating:**

The preliminary rating reflects the strong technical novelty and solid experimental validation of the proposed DiffTAC framework. The use of time-conditioned latent diffusion and the Integrated Attention Block is innovative and well motivated, and the results demonstrate clear improvements in image quality and temporal consistency across multiple datasets. However, the absence of experiments evaluating clinical utility limits the assessment of the method’s real-world impact. Addressing this limitation would likely justify a higher rating.

**Questions To Address In The Rebuttal:**

Can the authors evaluate the clinical utility of the generated images? While the reported image quality metrics indicate high fidelity, image quality alone may not be sufficient to demonstrate clinical usability in real-world medical settings.

---

> ### Author Response · Authors · 2026-01-20
> **Authors Responses to the Reviewer EZnh**
>
> Comment: Can the authors evaluate the clinical utility of the generated images? While the reported image quality metrics indicate high fidelity, image quality alone may not be sufficient to demonstrate clinical usability in real-world medical settings.
>
> Answer: We thank the reviewer for this important question. While our current evaluation focuses on image fidelity and temporal consistency, DiffTAC is motivated by clinically relevant constraints in cardiac MRI acquisition. Achieving high temporal-resolution cine MRI is challenging due to prolonged scan times and breath-hold limitations, which often limit the number of frames acquired per cardiac cycle. By enabling acquisition-free temporal super-resolution, DiffTAC provides temporally dense cine sequences without increasing scan duration or patient burden.
> Such high-temporal-resolution reconstructions can support more accurate analysis of cardiac motion, improve visualization of rapid physiological events, and benefit downstream applications such as myocardial motion assessment and fluid–structure interaction (FSI) modeling, where smooth and continuous ventricular motion is critical. In addition, temporally coherent reconstructions can facilitate digital twin–based cardiac simulations, enabling the study of disease-specific motion patterns under different physiological conditions. A comprehensive clinical validation, including expert assessment or task-based evaluation, is an important direction for future work and is beyond the scope of this study. In response to this, we moved the Potential Applications (Section 5) sections from the Appendix into the main body of the paper. We also moved the Discussion (Section 4) section from the Appendix into the main body of the paper, as the page limit has been increased to 12.

---

### Official Review · Reviewer_cMuM · 2026-01-06

**Confidence:** 4
**Preliminary Rating:** 4
**Final Rating:** 4

**Summary:**

This paper proposes a diffusion-based framework DiffTAC for synthesizing cardiac frames between end-diastole (ED) and end-systole (ES). To effectively combine temporal and anatomical information, the authors introduce the Integrated Attention Block. DiffTAC is evaluated on two public cardiac MRI (CMR) datasets.

**Strengths:**

1. They proposed the learnable sinusoidal embedding to model time as continuous variable.

2. Integrated Attention Block is able to combine both temporal and anatomical information.

3. Validate on two public datasets.

**Weaknesses:**

1. The experimental comparison lacks strong deep learning baselines or recent diffusion-based methods for temporal reconstruction.

2. The font sizes in Figure 2 appear inconsistent and should be unified for better readability.

**Detailed Comments:**

1. Add more comparison methods.
2. Make the font size consistent.

**Justification Of Final Rating:**

The authors have adequately addressed all of my questions and have additionally included further comparative experiments. I appreciate their efforts in improving the manuscript, and I am happy to recommend acceptance of this paper.

**Justification Of The Preliminary Rating:**

My preliminary rating is based on the novel method design for synthesizing cardiac frames under latent diffusion model. While the experimental comparison could be further strengthened, the core idea of temporally conditioned latent diffusion with anatomical anchoring is technically reasonable.

**Questions To Address In The Rebuttal:**

The manuscript would benefit from adding more experimental comparisons with existing diffusion methods if possible.

---

> ### Author Response · Authors · 2026-01-20
> **Authors Responses to the Reviewer cMuM**
>
> Comment 1: Add more comparison methods.
>
> Author Response: We thank the reviewer for the suggestion. In response, we have expanded the experimental comparison by adding two learning-based methods, UVI-Net and TSSC-Net, which cover the learning based methods. The results are reported in Table 2, and a dedicated discussion is provided in Section 3.5 (Comparison with State of the Art). For fairness, both methods are reproduced using their official codebases and evaluated under the same data splits and seed as DiffTAC. We believe these additions provide a stronger context for the proposed method and address the reviewer’s concern.
>
> Comment 2: Make the font size consistent.
>
> Author Response: We thank the reviewer for this suggestion. We have increased the font size in Figure 2.
>
> Also, we moved the Discussion (Section 4) and Potential Applications (Section 5) sections from the Appendix into the main body of the paper, as the page limit has been increased to 12.

---

### Official Review · Reviewer_ygSQ · 2026-01-09

**Confidence:** 5
**Preliminary Rating:** 2
**Final Rating:** 4

**Summary:**

The authors introduce DiffTAC, a latent diffusion framework for synthesizing intermediate cardiac phases by explicitly conditioning on time. The model leverages end-diastolic (ED) and end-systolic (ES) frames as anatomical anchors to guide the generation of temporally smooth cardiac motion. Quantitative evaluations on the Sunnybrook and ACDC datasets demonstrate that DiffTAC achieves improved temporal consistency and visual realism compared to classical interpolation and optical-flow–based methods.

**Strengths:**

The primary strength of the paper lies in its architectural contributions. The authors propose a latent diffusion model that incorporates a learnable sinusoidal embedding to encode time, enabling explicit temporal conditioning. The model further introduces Integrated Attention Blocks designed to jointly capture spatial and temporal information. In addition, the use of end-diastolic (ED) and end-systolic (ES) frames as anatomical anchors helps guide the generation process and improve temporal coherence.

**Weaknesses:**

While the paper presents strong architectural innovations, a key limitation lies in the benchmarking strategy. Although several relevant methods are discussed in the related work section, these approaches are not included in the quantitative or qualitative comparisons with DiffTAC. As a result, the evaluation primarily contrasts a relatively heavy, learning-based architecture against conventional interpolation or optical-flow methods, which may not provide a fully balanced or comprehensive assessment of the proposed model’s advantages.

**Detailed Comments:**

Major Points:

(1) Please clarify how the end-systolic (ES) and end-diastolic (ED) frames are incorporated during training. While the ED–ES pairing is mentioned, the exact role of each frame in conditioning or supervision is not entirely clear.

(2) The performance of linear spline interpolation and optical-flow methods drops substantially on the ACDC dataset compared to Sunnybrook. While some dataset-dependent variation is expected, it would be helpful to provide an explanation for why the degradation is so pronounced.

(3) Please consider benchmarking the proposed method against comparable learning-based diffusion models, such as TSSC-Net or UVI-Net. Although the proposed approach may not necessarily outperform these methods, including such comparisons would provide valuable context for readers and strengthen the evaluation.

Minor Points:

(1) On page 5, I think $z_{es}$ should be $z_{es} = \varepsilon (x_{\tau_{es})}$, instead of $T$.

(2) Is the subject-wise train–test split stratified by diagnostic category?

**Justification Of Final Rating:**

The authors addressed all my concerns by providing the necessary baselines and comparisons. They have also addressed all conceptual queries I had.  Their architectural contribution is the primary reason for the rating.

**Justification Of The Preliminary Rating:**

Overall, the paper presents a strong architectural contribution, but the evaluation would benefit from more comprehensive and appropriate benchmarking. I would be open to revising my score if the method is evaluated more fairly against other learning-based approaches.

**Questions To Address In The Rebuttal:**

I would like all major and minor points to be addressed.

---

> ### Author Response · Authors · 2026-01-20
> **Authors Responses to the Reviewer ygSQ**
>
> Comment 1: Please clarify how the end-systolic (ES) and end-diastolic (ED) frames are incorporated during training. While the ED–ES pairing is mentioned, the exact role of each frame in conditioning or supervision is not entirely clear.
>
> Answer: We thank the reviewer for this question. During training, the end-diastolic (ED) and end-systolic (ES) frames are not used as supervision targets, but are instead encoded using a frozen VAE and concatenated as latent features with the noisy latent of the target intermediate frame. These concatenated latents are fed into the diffusion U-Net, allowing the model to learn view-specific and anatomy-specific feature representations directly from the conditioning inputs. Supervision is applied only through noise prediction on the intermediate latent, and the model does not attempt to reconstruct ED or ES themselves.
> In addition to anchoring cardiac motion, conditioning on ED and ES constrains the generation to the appropriate anatomical feature space. Cardiac MRI can be acquired from different views (e.g., 2-chamber, 4-chamber, short-axis), and without ED/ES conditioning, the model could generate intermediate frames that are anatomically inconsistent or belong to a different view. By providing ED and ES latents, the model is guided to synthesize cine sequences that remain consistent with the given input anatomy and imaging plane, rather than producing unconstrained or random motion patterns. Temporal position is specified independently through a learnable sinusoidal embedding injected via cross-attention.
>
> We added one paragraph regarding this in the Discussion section (Section 4). We bring the Discussion and Potential Applications (Section 5) sections to the main part of the paper from the Appendix as the page limit is now increased to 12.
>
> Comment 2: The performance of linear spline interpolation and optical-flow methods drops substantially on the ACDC dataset compared to Sunnybrook. While some dataset-dependent variation is expected, it would be helpful to provide an explanation for why the degradation is so pronounced.
>
> Answer: We thank the reviewer for raising this point. The pronounced degradation of linear, spline, and optical-flow methods on ACDC can be attributed to several dataset characteristics that jointly increase interpolation difficulty. Compared to Sunnybrook, ACDC provides fewer frames per cardiac cycle, resulting in larger inter-frame motion between adjacent time points. This reduces the effectiveness of classical interpolation and optical-flow methods, which rely on small and smooth temporal displacements. In addition, ACDC volumes typically have larger slice thickness and greater inter-subject anatomical and pathological variability, further increasing motion ambiguity and reducing the reliability of motion estimation. Together, these factors explain the substantial drop in PSNR, SSIM, and LPIPS observed in Table 1. In contrast, Sunnybrook sequences are more densely sampled in time, where such methods degrade less severely.
>
> Comment 3: Please consider benchmarking the proposed method against comparable learning-based diffusion models, such as TSSC-Net or UVI-Net. Although the proposed approach may not necessarily outperform these methods, including such comparisons would provide valuable context for readers and strengthen the evaluation.
>
> Answer:
> We thank the reviewer for this valuable suggestion. In response, we have added explicit comparisons with learning-based methods by including UVI-Net and TSSC-Net in the revised evaluation. These results are reported in Table 2, and we have added a dedicated subsection, Section 3.5 (Comparison with State of the Art), to discuss the findings in detail.
> For a fair comparison, we reproduce both UVI-Net and TSSC-Net using their official codebases and evaluate them under the same data splits and seed as DiffTAC. This ensures consistency across all methods. We believe these additions strengthen the experimental evaluation and address the reviewer’s concern by situating DiffTAC more clearly with respect to existing learning-based and diffusion-based approaches.
>
> Comment 4: Minor correction
>
> Answer: We thank the reviewer for this comment. We have corrected the symbols in page 5.
>
> Comment 5: Is the subject-wise train–test split stratified by diagnostic category?
>
> Answer: We thank the reviewer for this question. We use an 80/20 train–test split at the subject level, where all frames from a given subject are assigned exclusively to either the training or the test set, ensuring no data leakage. Each subject contributes a complete cardiac sequence, and no frames are shared across splits. This evaluation protocol measures generalization to unseen subjects rather than unseen frames.
>
> All the changes in the paper are highlighted in red.

---

### Author Response · Authors · 2026-01-21
**Author Responses**

We have revised the manuscript in accordance with the reviewers' comments.  We wish to thank the reviewers for their meticulous critique that has led to an improved manuscript. Please note that the changes/revisions in the manuscript are highlighted in red color.

---

### Author Rebuttal · Authors · 2026-01-21

**Rebuttal:**

We have revised the manuscript in accordance with the reviewers’ comments and sincerely thank the reviewers for their meticulous and constructive critique, which has led to a significantly improved manuscript. All changes and revisions have been highlighted in red for ease of review. In addition, following the increase in the page limit to 12 pages, we have moved the Discussion (Section 4) and Potential Applications (Section 5) from the Appendix to the main body of the paper. We believe that the inclusion of these sections in the main manuscript strengthens the presentation and provides clearer context, deeper insight, and greater relevance of the results.

**Supporting Material:**

/attachment/a078a18095c03d1fb13a876a1fdadec2c3f6aaf3.pdf

---

### Meta-Review · Area_Chair_N6Uv · 2026-02-06

**Recommendation:** Accept (Poster)
**Confidence:** 4

**Metareview:**

The paper presents an approach to temporal super-resolution in cardiac MRI. The proposed diffusion-based temporal conditioning framework demonstrates clear advantages over existing methods, and the experimental results sufficiently support the claims. We therefore recommend acceptance.

---

### Decision · Program_Chairs · 2026-02-13

Accept (Poster)